# Diversity and evolution of the vertebrate chemoreceptor gene repertoire

**Maxime Policarpo** [1] ✉, **Maude W. Baldwin**[2], **Didier Casane**[3,4] & **Walter Salzburger** [1] ✉

Chemoreception – the ability to smell and taste – is an essential sensory modality of most animals. The number and type of chemical stimuli that animals can perceive depends primarily on the diversity of chemoreceptors they possess and express. In vertebrates, six families of G protein-coupled receptors form the core of their chemosensory system, the olfactory/pheromone receptor gene families *OR*, *TAAR*, *V1R* and *V2R*, and the taste receptors *T1R* and *T2R*. Here, we study the vertebrate chemoreceptor gene repertoire and its evolutionary history. Through the examination of 1,527 vertebrate genomes, we uncover substantial differences in the number and composition of chemoreceptors across vertebrates. We show that the chemoreceptor gene families are co-evolving, highly dynamic, and characterized by lineage-specific expansions (for example, *OR* in tetrapods; *TAAR*, *T1R* in teleosts; *V1R* in mammals; *V2R*, *T2R* in amphibians) and losses. Overall, amphibians, followed by mammals, are the vertebrate clades with the largest chemoreceptor repertoires. While marine tetrapods feature a convergent reduction of chemoreceptor numbers, the number of *OR* genes correlates with habitat in mammals and birds and with migratory behavior in birds, and the taste receptor repertoire correlates with diet in mammals and with aquatic environment in fish.

The survival of animals depends heavily on their ability to perceive their surroundings, for example, to orient themselves, to navigate through the environment, to find food, to escape from predators, and to identify and select mating partners[1–3]. These vital tasks are typically achieved by one or several of their sensory systems[3,4]. Different sensory modalities exist in animals that allow them to detect and interpret external or self-induced cues: photoreception for the detection of light, electroreception for the detection of electric signals, magnetoreception for the detection of magnetic fields, and chemoreception for the detection of chemical cues.

The chemosensory system combines the senses of smell (olfaction) and taste (gustation). In vertebrates, four multigene families are primarily responsible for olfaction: olfactory receptor (*OR*) genes[5], trace amine-associated receptor (*TAAR*) genes[6], and the vomeronasal receptor gene families *V1R* and *V2R*[7–9]. In tetrapods, *OR* and *TAAR* genes are primarily expressed in olfactory sensory neurons in the main olfactory epithelium. *V1R* and *V2R* genes are expressed in the sensory epithelium of the vomeronasal organ in tetrapods (except in amphibian, where *V1R* and a subset of *V2R* genes are expressed in the main olfactory epithelium)[10], while in cartilaginous and ray-finned fishes these genes – often referred to as *ORA* and *OlfC* in these clades – are expressed in the main olfactory epithelium[11,12]. Gustation, on the other hand, is achieved through the taste receptor gene families *T1R* (sweet and umami taste receptors) and *T2R* (bitter taste receptors), which are expressed in taste buds[13]. Overall, the range of molecules that can be recognized by a species depends in large part on the richness of the

[1]Zoological Institute, Department of Environmental Sciences, University of Basel, Basel, Switzerland. [2]Evolution of Sensory Systems Research Group, Max Planck Institute for Biological Intelligence, Seewiesen, Germany. [3]Université Paris-Saclay, CNRS, IRD, UMR Évolution, Génomes, Comportement et Écologie, Gif-sur-Yvette, France. [4]Université Paris Cité, UFR Sciences du Vivant, Paris, France. ✉ e-mail: maxime.policarpo@unibas.ch; walter.salzburger@unibas.ch

chemoreceptor gene repertoire[11,14]. Like the visual opsin genes that are at the core of the visual sensory system, the olfactory and gustatory receptor genes encode for G protein-coupled receptors (GPCRs)[4]. However, unlike the visual opsin genes, which are well characterized in vertebrates[15–17], the extent of the chemoreceptor gene repertoire as well as the evolutionary history of the different chemoreceptor gene families are only known for selected species or clades. To date, no overarching examination of the chemoreceptor gene repertoire exists across vertebrates, which is largely due to the sheer size of some of the chemoreceptor gene families and the application of different gene mining methodologies in previous studies, hampering comparisons between species and evolutionary lineages.

Here we examine the dynamics of chemoreceptor gene evolution across vertebrates. Making use of a newly developed computational pipeline, we mine 1527 vertebrate genomes for the six chemoreceptors gene families (*OR*, *TAAR*, *V1R*, *V2R*, *T1R* and *T2R*) in order to characterize the evolutionary history and diversification of these genes at a large scale. In addition, we test for associations between the chemoreceptor gene repertoires and eco-morphological proxies in the three largest vertebrate clades, ray-finned fishes, mammals and birds.

## Results

### Characterization of the vertebrate chemoreceptor gene repertoire

By applying a standardized procedure to detect chemoreceptor genes in 2210 vertebrate genome assemblies (and examining two datasets with different quality thresholds, at 80% and 90% complete BUSCO genes, retaining 1531 and 1180 genomes, respectively; Supplementary Table 1 and Supplementary Figs. 1–5), we found that the number of chemoreceptor genes is extremely variable across vertebrates (Figs. 1 and 2). Within olfactory and pheromone receptors (*OR*, *TAAR*, *V1R* and *V2R* genes combined), amphibians had the highest number of complete (that is, with a complete coding sequence [CDS]) genes (mean: 1060.3; minimum: 781; maximum: 1717; 20 species examined), followed by turtles (966.7; 258 to 1,716; 26 species), mammals (852.1; 33 to 2514; 440 species), lepidosaurs (539.7; 56 to 1035; 53 species), crocodiles (326; 16 to 743; 4 species), ray-finned fishes (238.6; 20 to 1388; 483 species), agnathans (121.4; 48 to 205; 5 species), birds (93.6; 4 to 1089; 488 species) and cartilaginous fishes (43.8; 20 to 62; 10 species). The lungfish (*Protopterus annectens*) and the coelacanth (*Latimeria chalumnae*) featured 989 and 280 complete olfactory receptor genes, respectively. Remarkably, with a mean number of 109.1 complete genes (minimum: 5; maximum: 268), amphibians also had the most extensive taste receptor gene repertoire (*T1R* and *T2R* genes combined) by far of any vertebrate clade. Except for the coelacanth (81 complete genes), the genomes of the other vertebrate groups contained less than one-fourth the number of complete taste receptor genes compared to amphibians: 4.1 (2 to 5) in cartilaginous fishes, 5.8 (0 to 24) in birds, 7.8 (5 to 12) in crocodiles, 9.1 (0 to 31) in ray-finned fishes, 11.8 (1 to 17) in turtles, 12.8 (0 to 61) in lepidosaurs, 23.4 (0 to 57) in mammals, and 24 in the lungfish. Amphibians thus emerge as the vertebrate group with the largest number of chemoreceptor genes per genome, followed by mammals and turtles (Fig. 2). The extended repertoire of chemoreceptor genes in amphibians is not primarily the result of whole genome duplication events in some of their representatives, as exemplified by the genus *Xenopus*: The diploid *X. tropicalis* had a very similar number of taste receptors (52) and an even higher number of olfactory receptors (1717) than the tetraploid species *X. laevis* (51 and 1265, respectively) and *X. borealis* (59 and 849, respectively).

Overall, we found positive correlations between the numbers of complete genes across the four different olfactory receptor gene families, suggesting that the evolution of these gene families has not been driven by compensatory gains and losses (Fig. 1b, Supplementary Fig. 6). We further tested for compensatory changes in repertoire size

of olfactory and taste receptors, as these can have an overlapping function, in particular in ray-finned fishes, where in some species taste buds are located across the body surface as well as in the oral cavity[18]. We again found a positive correlation between the number of olfactory receptor genes and the number of taste receptor genes in ray-finned fishes, mammals and birds (Fig. 1b, Supplementary Figs. 6 and 7), suggesting that these sensory modalities also evolve concertedly and that their evolution may be driven by the same environmental factors and/or life history traits. In addition, we found moderately positive correlations between the number of complete or total (sum of complete, pseudogenes and truncated genes) genes in each chemoreceptor family and genome size (Supplementary Figs. 8 and 9). Taking genome size into account, the numbers of complete genes were still correlated across chemoreceptor families (Supplementary Table 2).

### Evolution of *OR* and *TAAR* genes in vertebrates

The mean number of *OR* genes per genome ranged from 8.7 in Chondrichthyes (6 to 13) to 953.4 in turtles (252 to 1698), with an overall mean across vertebrates of 339.1 (Fig. 2a). The species with the highest number of complete *OR* genes was the short-beaked echidna (*Tachyglossus aculeatus*: *n* = 2399), closely followed by the Asian (*Elephas maximus indicus*: *n* = 2331) and the African elephant (*Loxodonta africana*: *n* = 2278) (see Supplementary Fig. 12 for details). We confirm previous results[19] that the *OR* gene repertoire of tetrapods is almost exclusively composed of genes belonging to the α-, β- and γ-subclades, while the α- and γ-subclades are either completely lacking or present in very low numbers in most ray-finned fishes (Supplementary Figs. 13–16). The *OR* gene repertoire of ray-finned fishes is, in turn, dominated by genes of the δ- and η-subclades, and to a lesser extent of the ζ-subclade. Whereas the coelacanth and the lungfish also featured genes of the ζ-subclade, these genes were lost in the evolutionary lineage leading to tetrapods. Genes of the η- and δ-subclades were also well represented in the coelacanth and lungfish genomes as well as in amphibians, but were lost before the most recent common ancestor (MRCA) of amniotes (Supplementary Fig. 16a). The ε-subclade was only retrieved in ray-finned fishes and amphibians, suggesting several independent losses of the ε-subclade, namely in cartilaginous fishes, in the coelacanth, in lungfishes, and before the MRCA of amniotes.

Trace amine-associated receptor (*TAAR*) genes were found in elevated numbers primarily in the genomes of ray-finned fishes, whereas their numbers were consistently low in tetrapods (Fig. 2b). The mean number of complete *TAAR* genes ranged from 1.6 (0 to 4) in birds to 51.7 in ray-finned fishes (3 to 497), with an overall mean of 19.7. The two Polypteriformes *Erpetoichthys calabaricus* and *Polypterus senegalus* featured by far the highest numbers of *TAAR* genes (*n* = 497 and *n* = 445, respectively), followed by the four-eyed sleeper *Bostrychus sinensis* (*n* = 307). On the other hand, some tetrapods completely lost their *TAAR* genes, such as the garter snake *Thamnophis elegans* (as opposed to 2 to 5 *TAAR* genes in all other snakes), the northern gundi (*Ctenodactylus gundi*), the four-striped grass rat (*Rhabdomys dilectus*) and several bird species. The comparatively large number of *TAAR* genes in Actinopterygii is largely due to an expansion of the B4-subclade, which is also present in coelacanth and lungfish, but absent in tetrapods (Supplementary Fig. 16b, Supplementary Figs. 17–20). We further found that tetrapods only have *TAAR* genes belonging to three subclades (A3, B1 and B3), one of which (B3) was lost in teleosts. *TAAR-like* genes, the sister subclade to all other *TAAR* genes and the only ones found in agnathans, are present in all vertebrate groups except amniotes (Supplementary Fig. 16b).

### Evolution of vertebrate vomeronasal receptors (*V1R* and *V2R*)

The mean number of complete *V1R* genes per genome ranged from 0.01 (0 to 1) in birds to 38 in mammals (0 to 276), with an overall mean of 13.1 (Fig. 2c). The number of complete *V1R* genes was particularly high in the platypus (*Ornithorhynchus anatinus*: *n* = 276) and in rodents

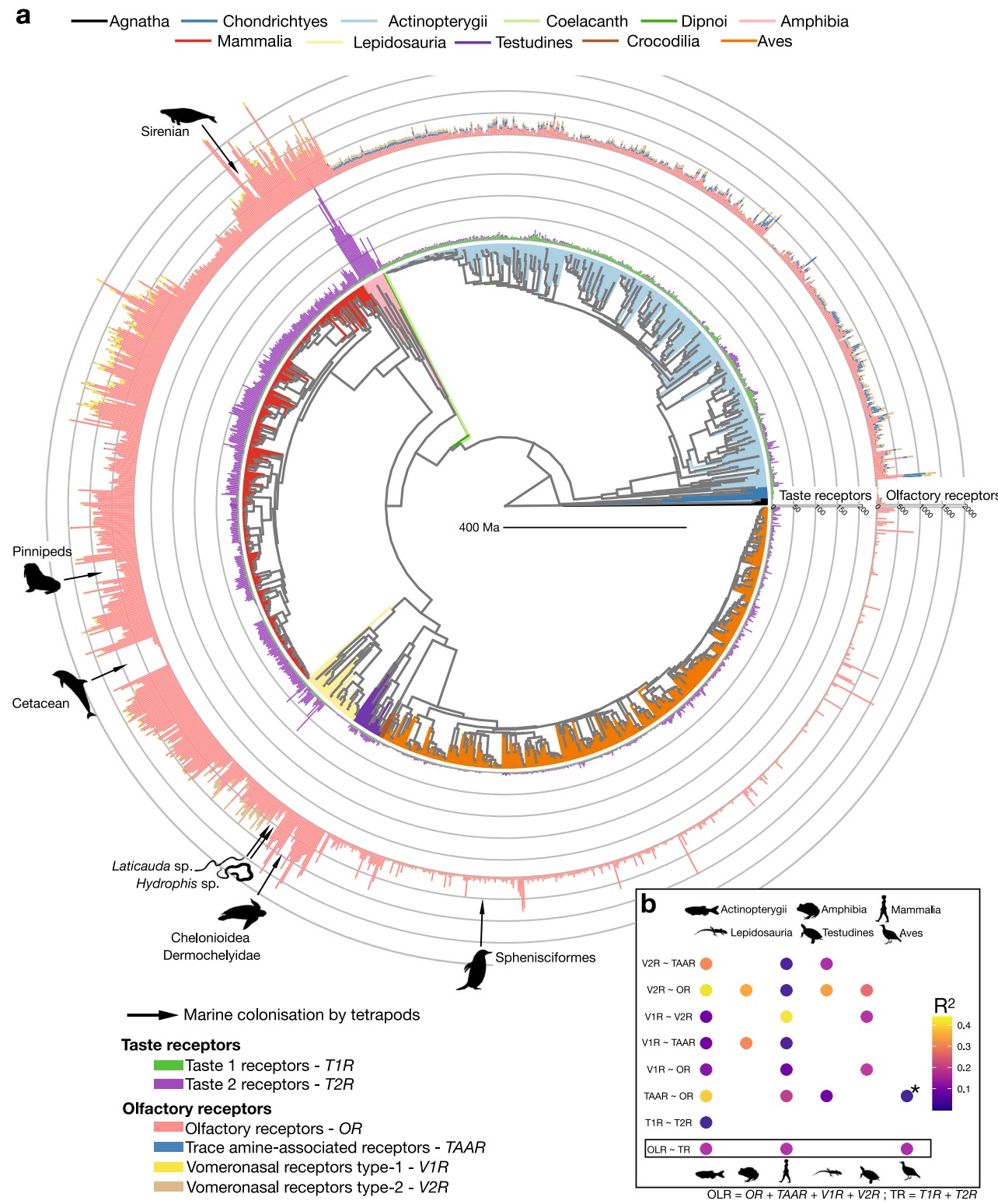

(*Mus musculus*: $n = 241$; *Arvicanthis niloticus*: $n = 187$) (Fig. 2c and Supplementary Fig. 21). The comparatively large *V1R* gene repertoire of mammals resulted primarily from an expansion of a single subclade (*V1R11*) in their ancestor, whereas in amphibians, which also have an extensive *V1R* gene repertoire, no major expansion in any particular subclade occurred, but they retained representatives of most subclades instead (Supplementary Figs. 22–25). It is of note that the lungfish is characterized by a rather large *V1R* gene repertoire with 163 complete genes, which is primarily due to an expansion of the *V1R9*

subclade. Most teleosts, on the other hand, retained only six *V1R* genes (*ORA1-ORA6*). We previously showed that this reduced repertoire is due to a series of gene losses before the MRCA of ray-finned fishes, followed by additional gene losses that occurred before the MRCA of teleosts and before the MRCA of clupeocephalans[20]. Contrary to what has previously been thought[21,22], we found that *V1R* genes are not entirely lacking from the genomes of crocodiles and birds, as one gene of the *V1R7* subclade was retained in some birds, and one gene each of the *V1R7* and *V1R10* subclades were retained in all crocodile species

**Fig. 1 | Co-evolution of chemoreceptor gene repertoires in vertebrates.**
**a** Phylogeny of 1532 vertebrate species, for which a genome assembly with more than 80% complete BUSCO genes was available (*Sceloporus occidentalis* is represented in the phylogeny but was excluded from the analysis; see Methods for details). The branches are colored according to the vertebrate (sub)class. The numbers of *OR, TAAR, V1R, V2R, T1R* and *T2R* genes for every species are shown as bars, color-coded as in the lower left panel. Independent marine colonization events by tetrapods (indicated by black arrows) are, for most parts, associated with decreases in chemoreceptor repertoire sizes. It is unknown whether the remaining genes in these species are functional in the context of chemoreception or used for other functions, as is the case for extranasal *OR* genes[115,116]. Phylogenies with full species names and sub-trees for each vertebrate (sub)class are available in

Supplementary Figs. 10 and 11, respectively. Animal silhouettes were obtained from PhyloPic.org. **b** Correlations between the number of complete genes of the different chemoreceptor families, or between the number of complete olfactory (OLR) and the number of complete taste receptors (TR) (BUSCO80 dataset; see Supplementary Fig. 7 for the results with the BUSCO90 dataset and when considering only chromosome-scale assemblies). Correlations were assessed using a two-sided pGLS. All correlations were positive. Circles indicate $P_{PGLS} < 0.05$ and are color-coded according to the pGLS $R^2$-values, absence of a circle indicates $P_{PGLS} > 0.05$. The association between *OR* and *TAAR* genes in birds (marked with an "*") is the only one that became non-significant with the BUSCO90 dataset. Samples sizes for chemoreceptor families correlations can be retrieved from Supplementary Table 1. Source data are provided as a Source Data file.

investigated. Despite the fact that bird *V1R7* genes all had an intact seven-transmembrane domain, these genes appear to be under relaxed selection, suggesting that they were retained by chance (Supplementary Fig. 26). We further document here that the substantial reduction of *V1R* genes in birds occurred gradually, with many gene losses occurring before the MRCA of amniotes, followed by a gradual loss of the remaining subclades (Supplementary Fig. 25a).

The mean number of *V2R* genes per species was found to be higher than that of *V1R* genes, ranging from 1.3 in turtles (0 to 3) to 140.6 in amphibians, with an overall mean of 23 across vertebrates (Fig. 2d and Supplementary Fig. 27). The V2R gene repertoire is particularly large in the lungfish (n = 493) as well as in two amphibians (*X. tropicalis*: n = 578, *X. laevis*: n = 388). *V2R* genes are separated in two main subclades, *V2RC* and *V2RD*. The *V2RC* subclade appears to have diversified before the MRCA of lungfishes and tetrapods, with several additional expansions detected in the lungfish (*V2RC6, V2RC7, V2RC8, V2RC9* and *V2RC10*), amphibians (mainly in *Xenopus*; *V2RC13*), mammals, and lepidosaurs (two independent expansions of *V2RC14*) (Supplementary Figs. 25b, 28–30). The *V2RD* subclade, on the other hand, has diversified before the MRCA of jawed vertebrates. However, most of these genes were lost in the lineage leading to tetrapods, suggesting that genes of this subclade are specialized to detect waterborne molecules. *V2R* genes are absent in most (263 out of 392) mammal species, as well as in all birds and crocodiles. In turtles, we found evidence for one to three complete *V2R* genes, except for nine turtle species – including the marine clade – that lacked *V2R* genes entirely.

In general, we found that the diversification of vomeronasal receptor genes is tightly connected with the evolution of the vomeronasal system itself. For example, the expansion and diversification of *V2RC* genes in the common ancestor of lungfishes and tetrapods coincides with the appearance of the vomeronasal organ[23]. In mammals, we found that *V1R* and *V2R* gene numbers are strongly correlated ($R^2 = 0.42$; $P < 2.2e−16$) and that groups known for their well-developed vomeronasal organs – such as rodents, lagomorphs, monotremes and marsupials – had comparatively larger *V1R* and *V2R* gene repertoires (Supplementary Figs. 6, 21 and 27). On the other hand, we provide evidence for a near-complete loss of *V1R* and *V2R* genes in turtles, crocodiles and birds, which either lack a vomeronasal system entirely or in which its presence is under debate[9]. To examine the co-evolution of pheromone receptors and the vomeronasal organ, we collected data on the presence/absence of an accessory olfactory bulb (AOB) in chiropterans[24–27]. We show that bats with an accessory olfactory bulb (AOB) have significantly more *V1R* genes than species without an AOB ($P_{PGLS} = 1.8e−5$; Supplementary Fig. 31). Furthermore, while PAML[28] did not detect a significant increase of the dN/dS on *V1R* genes of species without an AOB, a significant sign of neutral evolution was retrieved on these genes using RELAX[29] (K = 0.12, P = 0), suggesting a recent relaxation of selection (Supplementary Fig. 31).

### Evolution of vertebrate taste receptors (*T1R* and *T2R*)
Contrary to the four olfactory receptor gene families, which emerged in or before the vertebrate ancestor, we did not find any taste receptor

gene in agnathans (Fig. 2e, f). This suggests either a secondary loss of *T1R* and *T2R* genes in jawless vertebrates, or – much more plausibly – an origin of taste receptors in the evolutionary lineage leading towards gnathostomes, possibly in association with the emergence of the jaw apparatus itself and the subsequent diversification in feeding strategies[30,31].

In contrast to all other chemoreceptor gene families, which are highly dynamic with respect to gene duplications and losses across vertebrates, the number of *T1R* genes is rather stable, in particular in tetrapods, which typically feature three *T1R* genes: the umami and sweet receptor subunits (*T1R1A* and *T1R2*, respectively) and their co-receptor (*T1R3*) (Fig. 2e). Birds and teleosts independently lost their *T1R2* genes. The genomes of Actinopterygii, on the other hand, contain an additional *T1R* subclade that was lost before the MRCA of Sarcopterygii, which we named *T1R1B* (Supplementary Figs. 32–35). Previous studies have treated this clade as part of the *T1R2* subclade[32,33]. However, our phylogeny suggests a possibly more complicated evolutionary scenario, and whether this clade is part of the *T1R1* or *T1R2* clade is not entirely clear. This ray-finned fish clade is more dynamic than other *T1R* subclades, resulting in ray-finned fishes having a greater number of *T1R* genes compared to all other vertebrate clades (Fig. 2e and Supplementary Figs. 32 and 34). The species with the highest number of complete *T1R* genes were the Chinese sleeper (*P. glenii*: n = 18) and the spinyhead croaker (*Collichthys lucidus*; n = 18), followed by the jewelled blenny (*Salarias fasciatus*; n = 16) (Fig. 2e). Several lineages completely lost their *T1R* genes, such as the genus *Xenopus*, cetaceans, pinnipeds, the marine turtle *Dermochelys coriacea*, most snakes (which also exhibited a reduced *T2R* gene repertoire with 0 to 2 complete genes), and two bird orders (Sphenisciformes [penguins] and Tinamiformes) (Supplementary Figs. 32 and 36).

It has previously been hypothesized (but was not formally tested) that the evolution of *T1R* genes in mammals was shaped by diet specializations[34,35]. In our analyses, we identified 17 independent losses of *T1R2* on carnivore branches within mammals, 6 on herbivores and 2 on omnivores (Fig. 3), resulting in more than half (63 in a total of 117) of the mammalian carnivore species lacking the sweet receptor subunit (*T1R2*). Using a combination of BayesTrait analysis and simulations, we found that, in mammals, carnivores were significantly more prone to lose *T1R2* compared to herbivores and omnivores ($P_{BayesTrait} = 6e−4$; Supplementary Tables 3 and 4, Supplementary Fig. 37), and experienced significantly more *T1R2* losses than what would be expected at random ($P_{Simulations} = 5e−4$; Fig. 3 and Supplementary Fig 38). This association holds true when removing pinnipeds and cetaceans, which may have lost their *T1R* genes in response to their transition to a marine lifestyle rather than their diet (as observed for other chemoreceptor genes). We would also like to note that, although the loss of *T1R2* genes is not reversible, a transition from carnivore to omnivore/herbivore diet[36] could potentially involve shifts in taste receptor function, as has been shown for hummingbirds[37]. No significant association was found between *T1R1* gene losses and diet preference in mammals (eight in carnivores, nine in herbivores, and five in omnivores) nor between *T1R3* gene losses and diet (six in carnivores, nine in

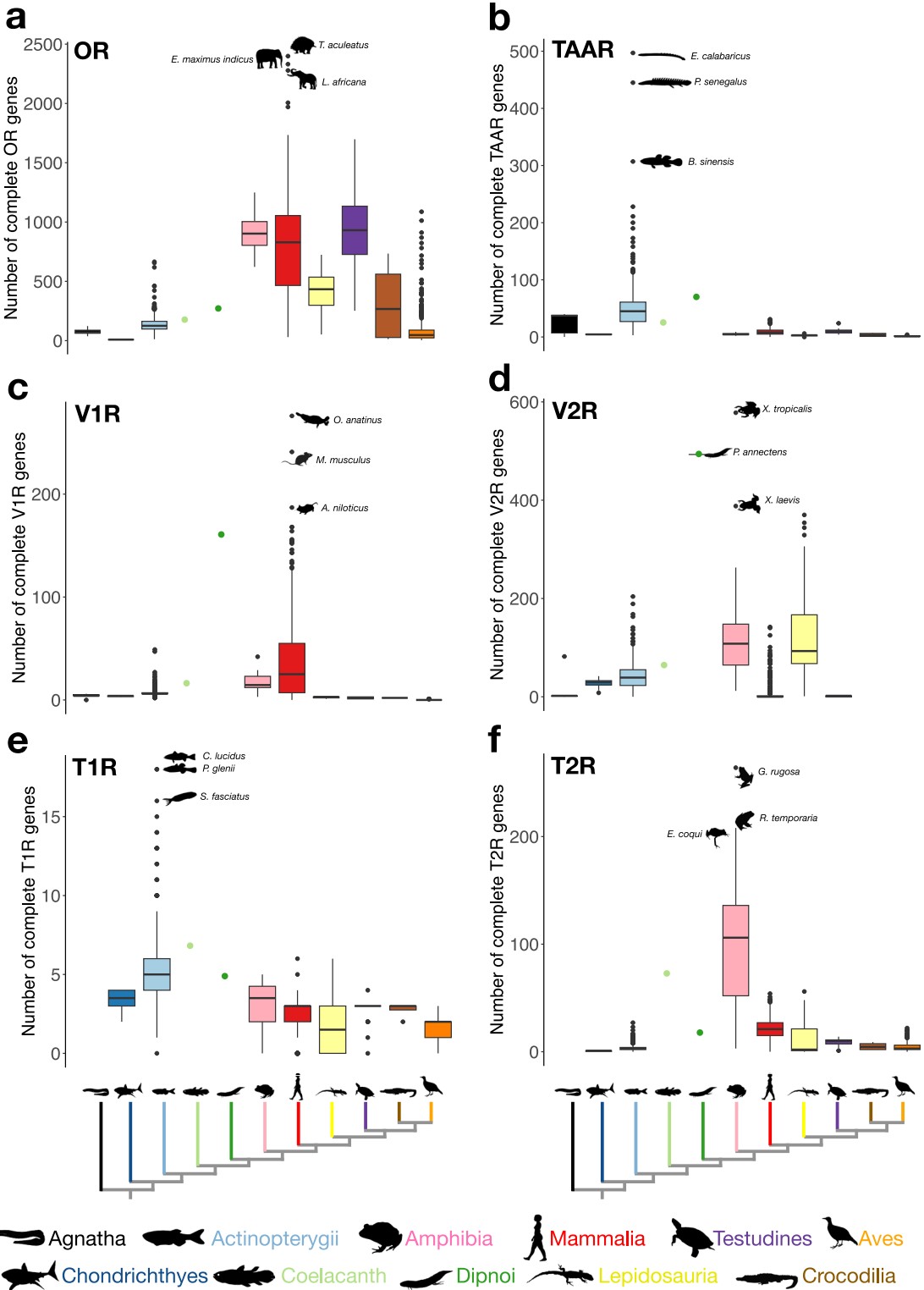

**Fig. 2 | Number of chemoreceptor genes in vertebrates.** For each vertebrate (sub) class (colored as in Fig. 1), the number of olfactory and taste receptor genes is shown as boxplots (first quartile −1.5 interquartile range; first quartile; mean; third quartile; third quartile +1.5 interquartile range; dots represent outliers) for the BUSCO80 dataset. For each chemoreceptor gene family, the names of the three species with the highest number of genes, and their silhouettes, are shown. **a** *OR* genes; **b** *TAAR* genes; **c** *V1R* genes; **d** *V2R* genes; **e** *T1R* genes; **f** *T2R* genes. The species with the highest number of complete olfactory receptor genes is

*Tachyglossus aculeatus* (2514) closely followed by *Elephas maximus indicus* (2383) and *Loxodonta africana* (2329), while the species with the highest number of complete taste receptor genes is *Glandirana rugosa* (268). Note that the high number of complete *OR* genes found in *Tachyglossus aculeatus* could potentially represent an artifact, as we also retrieved an unusually high number (nearly 9000) of incomplete genes in this species (Supplementary Data 1). Samples sizes for each vertebrate (sub)class can be retrieved from Supplementary Table 1 ("Nb of species >80% BUSCO"). Source data are provided as a Source Data file.

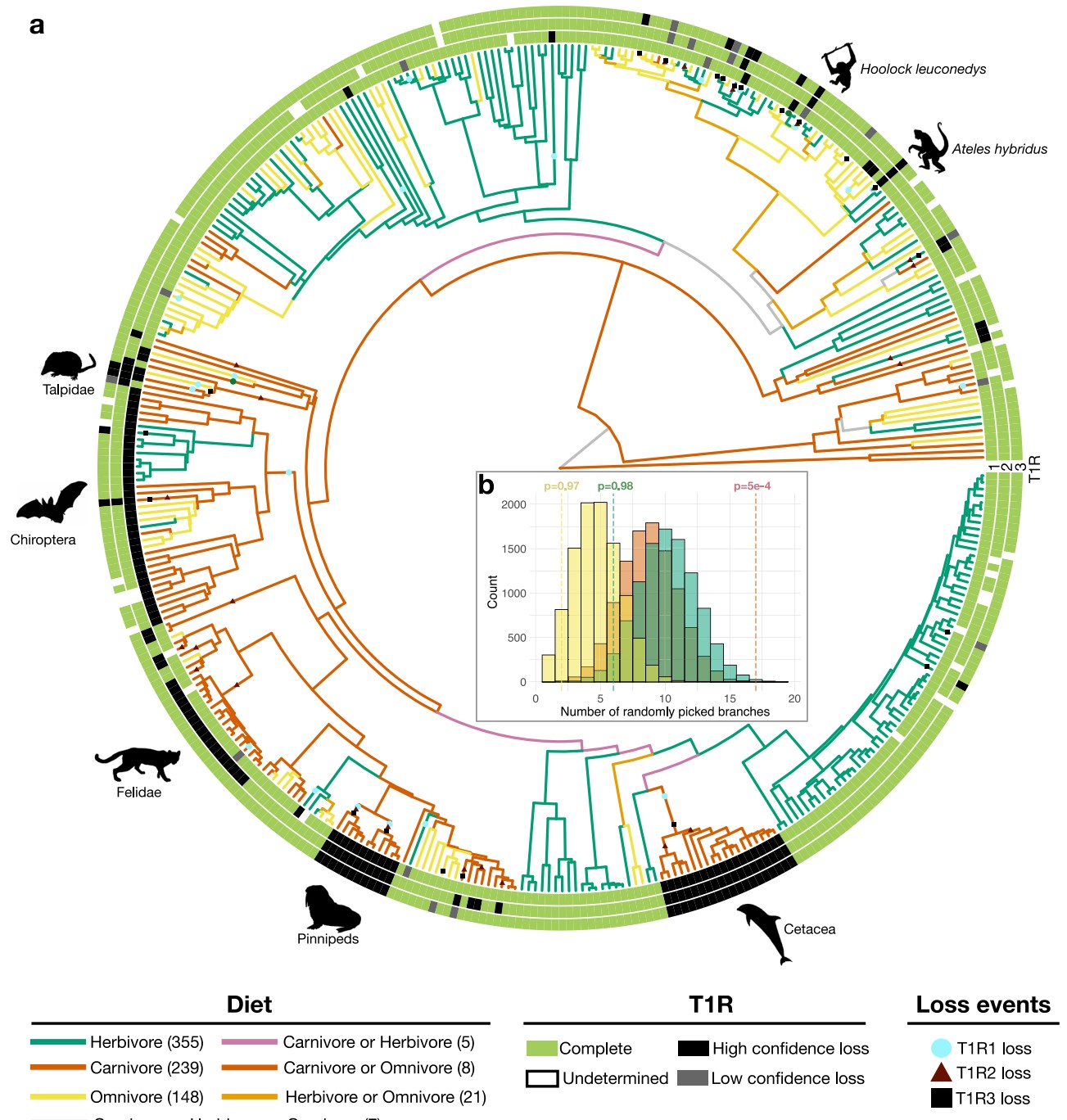

**Fig. 3 | Repeated loss of *T1R2* in carnivore mammals. a** Phylogeny of mammals, for which a genome assembly with more than 80% complete BUSCO genes was available (392 species). Terminal branches are color-coded according to the diet preference taken from the MammalDiet database[43]. Diet preferences of internal branches were inferred with PastML. The status of each gene (*T1R1, T1R2, T1R3*) in each species is indicated (according to the four categories shown; see Methods for details). *T1R* loss events, inferred by shared loss-of-function mutations across species, are indicated on the respective branches. Large clades with *T1R* losses, or individual species that have lost all *T1R* genes, are highlighted with a silhouette. **b** Simulation result where *T1R2* genes were randomly pseudogenized in the

mammalian tree. The histogram represents the results of the simulations (with the x-axis representing the number of randomly drawn branches in the simulations) and the dashed lines represent the observed number of independent *T1R2* loss per diet group (same color code as in the phylogeny). The *P*-value reported above each dashed line correspond to the number of simulations where the same or a greater number of independent *T1R2* losses occurred than observed for the same branch category (carnivore, omnivore or herbivore), divided by the total number of simulations (10,000). All simulation results for *T1R1, T1R2* and *T1R3* are shown in Supplementary Fig. 38. Source data are provided as a Source Data file.

herbivores, and six in omnivores). Most species that lack a complete *T1R3* gene lost the *T1R1* and/or the *T1R2* subunits (43 species, 86%), while 7 species (14%) still have intact *T1R1* and *T1R2* subunits. In sirenians, which – just like pinnipeds and cetaceans – experienced a

massive loss of olfactory receptors and of *T2R* (see below), the three *T1R* genes were intact and seem to be under negative selection (Supplementary Figs. 39–41). However, a small but significant increase of the ω ratio (that is, dN/dS of *T1R1* and *T1R3* genes in sirenians

compared to terrestrial species could be a sign of an episodic event of positive selection acting on these genes (Supplementary Figs. 39–41).

Whereas the *T2R* gene repertoires of ray-finned fishes and the lungfish are relatively small, a diversification and expansion of this gene family occurred in the lineage leading to tetrapods, followed by subsequent expansions of two subclades in amphibians (*T2RE3* and *T2RE5*) and two other subclades in mammals (*T2RE12* and *T2RE13*) (Supplementary Figs. 35b, 36 and 42–44). With a mean number of 106.2 complete genes, amphibians had a remarkably large *T2R* gene repertoire (ranging from 3 in *Geotrypetes seraphini* to 264 in *Glandirana rugosa*), followed by mammals with a mean number of 21.1 complete *T2R* genes (0 to 54) (Fig. 2e). Some tetrapods completely lost their *T2R* gene repertoire secondarily, such as Sphenisciformes and some cetaceans. With the identification of a single *T2R* gene in most cartilaginous fishes, we can reject the prevailing view of an origin of *T2R* genes in bony fishes[38,39].

Similar to what has been suggested for the evolution of *T1R* in mammals, it has previously been proposed that the vertebrate *T2R* gene repertoire evolved in response to diet preferences, with herbivores having more *T2R* genes than carnivores in order to detect toxic compounds in plants[40]. To examine this hypothesis, we retrieved the diet preferences of ray-finned fishes[41], birds[42] and mammals[43]. We found a correlation between the number of complete *T2R* genes and diet categories in mammals ($P_{BUSCO80} = 0.007$; $P_{BUSCO90} = 0.013$). Contrary to previous assumptions[40], we found that omnivores, not herbivores, had the highest *T2R* copy number (mean$_{BUSCO80} = 26.4$; mean$_{BUSCO90} = 26.6$), then followed by herbivores (mean$_{BUSCO80} = 22.9$; mean$_{BUSCO90} = 22.7$), and then carnivores (mean$_{BUSCO80} = 14.1$; mean$_{BUSCO90} = 14.4$). This correlation holds true when cetaceans and pinnipeds are removed at the 80% BUSCO completeness threshold (but not at 90%). Also, this association between diet and the *T2R* repertoire size is not true for ray-finned fishes ($P_{BUSCO80} = 0.47$; $P_{BUSCO90} = 0.25$) nor for birds and crocodiles ($P_{BUSCO80} = 0.1$; $P_{BUSCO90} = 0.15$) (Fig. 4, Supplementary Fig. 45). It should also be emphasized that amphibians, which are all carnivores, have an extensive *T2R* repertoire (Fig. 2f).

**The eco-morphology of chemoreceptor evolution in vertebrates**
We examined correlations between ecological traits as well as diet preferences (from existing databases[41–45]) and olfactory gene repertoire sizes (this study) across the three vertebrate groups with most genome assemblies available: mammals, birds and ray-finned fishes. Our analyses revealed a strong association between the number of complete *OR* genes and habitat in mammals (marine, ground level, scansorial, arboreal or aerial; Table 1) as well as between their number of complete *TAAR* genes and their habitat. There was also a correlation between the number of complete *OR* genes and habitat in birds using our BUSCO filtered datasets, but this became non-significant when taking only chromosome-scale assemblies (Table 1). We also detected a correlation between the number of complete *OR* genes and the migratory behavior in birds using both BUSCO datasets, with non-migratory species having fewer *OR* genes than migratory ones, but again, this correlation is no longer significant when using only chromosome-scale assemblies (Table 1). Finally, in birds, there was a robust association between their primary lifestyle (aerial, terrestrial, aquatic or insessorial, i.e., species spending much of the time perching above the ground) and the number of complete *OR* genes (Table 1). We then assessed possible links between the taste receptor repertoire and ecological parameters. In Actinopterygii, the number of *T1R* genes was associated with the primary aquatic habitat (fresh, brackish or salt water, or combinations thereof; Table 1). In birds, we found a correlation between the number of *T2R* genes and the migratory behavior (Table 1, Fig. 4 and Supplementary Fig. 46).

The transition of tetrapods towards a marine lifestyle appears to have a particularly strong impact on the number of chemoreceptor genes in a given genome[46–50]. We consistently found a reduction of chemoreceptor genes across marine groups (cetaceans, pinnipeds and sirenians in mammals; penguins in birds; marine turtles; marine snakes of the genera *Hydrophis* and *Laticauda*) (Figs. 1 and 4). In these marine clades, the remaining complete *OR* genes do not seem to have experienced accelerated evolution (either by positive selection or relaxation of selection). Indeed, a mean of only 23% of remaining *OR* genes had a significantly accelerated evolution in cetacean compared to closely related terrestrial species. Other marine tetrapod clades had an even lower proportion of *OR* genes with a significant higher ω ratio compared to close terrestrial species (mean of 9.9%, 11.7%, 1.9%, 2.5% and 2% in pinnipeds, sirenian, sphenisciformes, sea snakes and sea turtles respectively, Supplementary Figs. 47–52).

Finally, we support the previously suggested association between the number of *OR* genes and the morphology of the olfactory organ[20,51,52]. More specifically, we found positive correlations between the number of complete *OR* genes and the relative size of the olfactory bulb in birds[53] (pGLS$_{BUSCO80}$: $R^2 = 0.11$, $P = 0.01$) and mammals[54] (pGLS$_{BUSCO80}$: $R^2 = 0.14$, $P = 0.02$), and between the number of complete *OR* as well as the number of complete *V2R* genes and the number of lamellae in the olfactory epithelium in ray-finned fishes (pGLS$_{BUSCO80}$: $R^2 = 0.17$, $P = 3.9e{-}5$ for *OR* genes; pGLS$_{BUSCO80}$: $R^2 = 0.07$, $P = 0.01$ for *V2R* genes; Supplementary Fig. 53). Interestingly, we found that, in mammals, the number of complete *TAAR* genes – although not prominent nor very dynamic – is also positively correlated with the relative olfactory bulb size (pGLS$_{BUSCO80}$: $R^2 = 0.56$, $P = 2.8e{-}7$). These associations with olfactory organ morphologies hold true when considering the BUSCO90 or chromosome-scale assemblies datasets (Supplementary Fig. 53).

Finally, our results support the correlation between the number of *OR* gene and the number of olfactory turbinals in mammals[55] (pGLS: $R^2 = 0.28$, $P = 9.4e{-}5$, Supplementary Fig. 5). Although different ecological parameters could also be associated with differences in olfactory and taste receptor expression, the lack of transcriptome data for olfactory epithelium and taste buds among closely related organisms complicates such investigations.

## Discussion
In this study, we characterize the chemoreceptor gene repertoires of vertebrates using a gene mining approach and applying it to 1527 vertebrate genome assemblies. We provide an updated nomenclature of vertebrate olfactory/pheromone and taste receptor genes based on extensive phylogenetic analyses across all vertebrate (sub)classes and chemoreceptor multigene families, reconstruct the dynamic evolution of vertebrate chemoreceptor genes, and identify ecological and morphological correlates of chemoreceptor evolution.

First of all, we show here that the sizes of the six chemoreceptor gene families – the olfactory and pheromone receptor gene families *OR*, *TAAR*, *V1R* and *V2R* and the taste receptors *T1R* and *T2R* – differ greatly among vertebrate species (Fig. 1). Likewise, we found substantial differences across vertebrate (sub)classes with respect to the total number of complete genes they possess from the six different chemoreceptor gene families (Fig. 2), and in the group-specific compositions of chemoreceptors (Supplementary Figs. 16, 25 and 35). Turtles, closely followed by amphibians and mammals, have the highest median numbers of complete *OR* genes per genome (Fig. 2a). In terms of the *OR* subclades, however, the genomes of aquatic and semi-aquatic vertebrate (sub)classes contain a greater *OR* subclade diversity compared to terrestrial (that is, amniote) lineages (Supplementary Fig. 16a). Ray-finned fishes (and above all Polypteriformes), together with the lungfish, stand out by their comparatively large number of *TAAR* genes (Fig. 2b); the genomes of ray-finned fishes also show the greatest *TAAR*-subclade diversity (Supplementary Fig. 16b). Mammals, followed by amphibians, have the highest numbers of the *V1R* vomeronasal chemoreceptor genes (Fig. 2c), while amphibians,

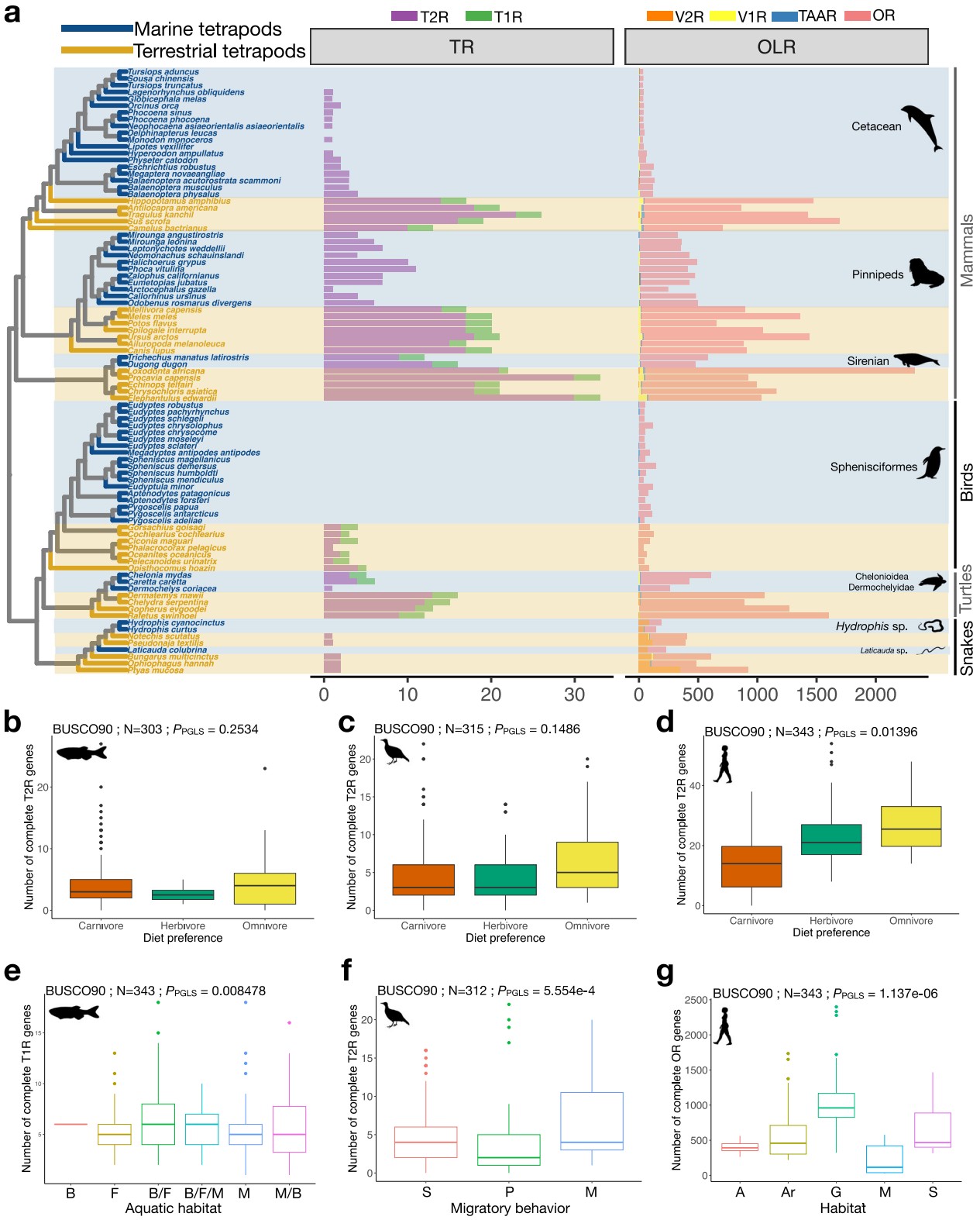

closely followed by lepidosaurs, have the highest numbers of the *V2R* vomeronasal receptors (Fig. 2d). The lungfish has a high number of both *V1Rs* and *V2Rs* (Fig. 2c, d), making it the vertebrate with the highest number of vomeronasal chemoreceptor genes. Aquatic and semi-aquatic vertebrate (sub)classes, in particular amphibians, have a much greater *V1R* subclade diversity than terrestrial ones (Supplementary Fig. 25a). With respect to *V2Rs*, the lungfish has the greatest subclade diversity, sharing the C-subclade with tetrapods and the

D-subclade with its aquatic ancestors (Supplementary Fig. 25b). The genomes of ray-finned fishes, together with the coelacanth and the lungfish, feature the highest numbers of the taste receptor genes *T1R* (Fig. 2e), whereas amphibian genomes contain the by far the largest *T2R* gene repertoire (Fig. 2f). Although the *T1R* subclade diversity is similar across vertebrate clades (Supplementary Fig. 35a), the *T2R* subclade diversity is much greater in the semi-terrestrial amphibians and in the terrestrial amniotes (Supplementary Fig. 35b). Overall,

**Fig. 4 | Ecology of chemoreceptor evolution in vertebrates. a** Phylogeny of marine tetrapods and closely related species, displaying the number of olfactory and taste receptors. Genes are color-coded according to chemoreceptor family. The names of the marine species and the associated branches in the phylogeny are colored in blue, while the names of non-marine species and the respective branches are colored in brown. Marine clades feature a reduction in the number of olfactory and taste receptor genes. All marine clades, except sirenians, have lost their *T1R* genes; *T2R* genes are completely lacking from the genomes of Sphenisciformes and some cetaceans. Association between the number of *T2R* genes and diet preferences in ray-finned fishes assessed with a two-sided pGLS test (**b**), in birds and crocodiles (**c**), and in mammals (**d**). pGLS *P*-values are reported above each boxplot (for the BUSCO90 dataset; for BUSCO80 results and chromosome-scale assemblies see Supplementary Fig. 45), N refers to the number of genomes used for the respective analysis. For mammals, we further tested the impact of the two carnivore marine clades on the pGLS results (Supplementary Fig. 45). **e**–**g** Two-sided pGLS test results between ecological parameters and the number of chemoreceptor

genes. Association between the number of complete *T1R* genes and the habitat in ray-finned fishes (**e**), the number of complete *T2R* genes and the migratory behavior in birds (**f**), and the number of complete *OR* genes and habitat in mammals (**g**). pGLS *P*-values are reported above each boxplot (for the BUSCO90 dataset, N refers to the number of genomes used for the respective analysis. Note that pGLS *P*-values are also significant with the BUSCO80 dataset or when considering only chromosome-scale assemblies (see Supplementary Fig. 46). Aquatic habitat in ray-finned fishes is coded as B...brackish, F...freshwater, M...marine (allowing for combinations); migratory behavior in birds is coded as S...sedentary, P...partially migratory, M...migratory; habitat in mammals is encoded according to their For-Strat.Value as M...marine, G...ground level including aquatic foraging, S...scansorial, Ar...arboreal, A...aerial (see Supplementary Data 1). Boxplots represent the first quartile −1.5; the interquartile range, the first quartile, the mean, the third quartile and the third quartile +1.5 interquartile range. Dots represent outliers. Source data are provided as a Source Data file.

amphibians turn out to be the clade with the largest chemoreceptor gene repertoire within vertebrates and it is reasonable to assume that the greater number of chemoreceptor genes and the greater representation of subclades of chemoreceptor gene families in amphibian genomes is due to their semi-aquatic (as larvae) / semi-terrestrial (as adults) lifestyle, requiring adaptations to both realms, in combination with their intermediate phylogenetic position between aquatic and primarily terrestrial vertebrate clades. That the number of olfactory receptor genes correlates with the number of taste receptors in many vertebrate clades (Supplementary Fig. 6) argues against compensatory mechanisms between olfaction and gustation and highlights the importance to jointly consider both chemoreceptor subtypes.

Our study also reveals morphological and ecological correlates of chemoreceptor evolution in vertebrates. We confirm that the olfactory receptor gene repertoire of ray-finned fishes is associated with the complexity of their olfactory epithelium[20]. Thus, in these species, increasing the number of neurons could allow the expression of more olfactory receptor genes, increasing the range of detectable odors. This is in contrast with what is observed in cartilaginous fishes, which harbor a high number of olfactory sensory neurons with a small repertoire of olfactory receptor genes, which could allow a higher expression of each receptor, hence increasing the sensitivity toward a restricted range of odorant molecules. Moreover, we confirm previous results[51,52] that the olfactory bulb size is positively correlated with the number of olfactory receptor genes in birds and mammals. Whether such associations are also true for other vertebrate clades, for which limited data with respect to their olfactory organ's sizes are available, is an open question. It is also unknown if the increase in the size of the olfactory epithelium, or the olfactory bulb size, has driven the expansion of olfactory receptor genes or vice versa, or if they co-evolved. We further uncover a strong correlation between the *OR* gene repertoire and habitat in mammals and birds, and between *OR* (and to some extent *T2R*) genes and migratory behavior in birds, and show that carnivorous mammals are more prone to *T1R2* (sweet receptor) gene losses than omnivorous or herbivorous ones.

The transition towards a marine lifestyle appears to have had a particularly strong impact on the chemoreceptor genes in tetrapods, with marine species generally featuring an impoverished repertoire (Figs. 1 and 4). For example, cetaceans and pinnipeds completely lack *T1R* genes. It has previously been suggested that the loss of *T1R* genes in these carnivorous marine mammals is due to the high sodium concentration in oceans[56,57]. However, we show here that *T1R* genes are still present in sirenians, which are herbivorous and marine mammals, casting doubts on this hypothesis and suggesting instead that *T1R* losses in marine mammals are associated with dietary adaptations and/or other ecological factors. In yet another aspect of convergent evolution between these evolutionary lineages, penguins have also lost their *T1R* (and *T2R*) genes. This has previously been associated with

their life in cold environments[50]. However, we found that the genomes of other representatives of cold-adapted tetrapods – such as the muskox (*Ovibos moschatus*) and the reindeer (*Rangifer tarandus*), and in particular the carnivore species snowy owl (*Bubo scandiacus*), arctic fox (*Vulpes lagopus*), and polar bear (*Ursus maritimus*) – contain *T1R*s and typically many *T2R*s. Thus, it remains hard to decipher why penguins completely lost their taste receptors.

In summary, we highlight the relevance of examining the six chemoreceptor families together in vertebrates, and provide novel insights into ecological factors driving the chemoreceptor repertoire evolution. Our dataset and gene mining procedure will be a valuable resource for future chemoreceptor studies, especially in the light of more and more genome assemblies becoming available.

## Methods
### Genome data
To download all vertebrate genomes available at the NCBI public database as of 31 July 2022, we used the program genome_updater with the options -T "7742" (corresponding to the taxonomic ID of vertebrates at NCBI), -d "refseq,genbank" (to browse both the RefSeq and Genbank databases), and -A 1 (to retain only one genome assembly per species). In total, we downloaded 2386 vertebrate genomes, of which 176 were removed as their assembly were described as "partial genome assembly" in NCBI, leading to a final dataset comprising 2210 vertebrate genomes (Agnatha: 5; Chondrichthyes: 14; Actinopterygii: 900; Dipnoi: 2; Coelacanth: 1; Amphibia: 32; Mammalia: 555; Lepidosauria: 66; Testudines: 27; Crocodilia: 4; Aves: 604). Three diploid genome assemblies were present in the dataset (*Neopelma chrysocephalum*: GCA_003984885.2; *Acinonyx jubatus*: GCA_003709585.1; *Tenualosa ilisha*: GCA_015244755.2). For these assemblies, we kept only scaffolds corresponding to the principal pseudohaplotype. Two haploid genome assemblies contained alternate loci scaffolds which were removed (*Homo sapiens*: GCA_000001405.29; *Danio rerio*: GCA_000002035.4).

### Phylogenies
To obtain a phylogenetic hypothesis for the vertebrate species included in this study, we merged available phylogenies for different vertebrate (sub)classes into a single tree. The phylogenetic tree for Agnatha was retrieved on TimeTree.org[58]. Phylogenetic trees for Amphibia, Mammalia, Aves, Lepidosauria and Chondrichthyes were downloaded from https://vertlife.org/[59–63]. We followed the suggestion by[64] and downloaded 1000 trees for each (sub)class and summarized these into a 50% majority-rule consensus tree using the sumtree.py script in the Dendropy package[65]. The phylogeny of Actinopterygii was obtained from https://fishtreeoflife.org/[66]. For Crocodilia and Testudines, we used previously published phylogenies[67,68]. For 161 taxa, species names had to be modified in order to match the ones from our genomic dataset (Supplementary Data 1; verified using https://www.

**Table 1 | Associations between ecological parameters and the number of chemoreceptor genes**

| (Sub)class | Response | Predictor | Dataset | N | λ | R² | P-value |
|---|---|---|---|---|---|---|---|
| Mammalia | OR | Habitat | BUSCO80 | 392 | 0.99 | 0.07 | 2e–5 |
| | | | BUSCO90 | 343 | 0.99 | 0.09 | 1e–6 |
| | | | Chromosome | 126 | 0.99 | 0.14 | 8e–4 |
| | TAAR | Habitat | BUSCO80 | 392 | 0.87 | 0.04 | 4e–3 |
| | | | BUSCO90 | 343 | 0.85 | 0.05 | 7e–4 |
| | | | Chromosome | 126 | 0.9 | 0.13 | 2e–3 |
| Aves | OR | Habitat | BUSCO80 | 467 | 0.19 | 0.04 | 2.5e–2 |
| | | | BUSCO90 | 312 | 1e–6 | 0.18 | 4.4e–9 |
| | | | Chromosome | 83 | 0.13 | 0.12 | 0.15 |
| | OR | Migratory behavior | BUSCO80 | 467 | 0.3 | 0.02 | 4.5e–3 |
| | | | BUSCO90 | 312 | 0.4 | 0.05 | 5e–4 |
| | | | Chromosome | 83 | 0.31 | 0.05 | 0.14 |
| | OR | Primary lifestyle | BUSCO80 | 467 | 0.31 | 0.04 | 2e–3 |
| | | | BUSCO90 | 312 | 0.42 | 0.03 | 0.07 |
| | | | Chromosome | 83 | 0.23 | 0.17 | 6e–3 |
| | T2R | Migratory behavior | BUSCO80 | 467 | 0.47 | 0.014 | 3.6e–2 |
| | | | BUSCO90 | 312 | 0.67 | 0.05 | 5.6e–4 |
| | | | Chromosome | 83 | 1e–6 | 0.09 | 0.02 |
| Actinopterygii | T1R | Aquatic habitat | BUSCO80 | 447 | 0.89 | 0.04 | 4.3e–3 |
| | | | BUSCO90 | 343 | 0.88 | 0.05 | 8.5e–3 |
| | | | Chromosome | 210 | 0.7 | 0.05 | 0.0498 |

To test for associations between the number of genes in each chemoreceptor subfamily (response) and ecological parameters (predictor) in mammals, birds and ray-finned fishes, we ran a two-sided pGLS with caper. The complete list of ecological parameters tested and all pGLS results are reported in Supplementary Table 7, Supplementary Data 1 and Supplementary Fig. 46. "Chromosome" dataset corresponds to pGLS performed by considering only chromosome-scale assemblies. N corresponds to the number of species included in the pGLS analysis. pGLS results (λ, R² and P-value) are indicated. Source data are provided as a Source Data file.

itis.gov/, https://www.marinespecies.org/ and https://avibase.bsc-eoc.org/). We also inferred the phylogenetic position of 59 species for which a genome was available, but which were not included in the available phylogenies, using genus information (Supplementary Data 1). Among the 2210 species with a genome available, a total of 222 species were excluded because they were hybrid or extinct species, or because it was not possible to infer their phylogenetic position. The different phylogenies were combined with the coelacanth and the two Dipnoi species with the bind.tree function (which can also bind a single tip into a tree) in ape v5.0[69], using the divergence times available from TimeTree.org[58], for a final tree containing 1988 vertebrate species.

**Genome completeness assessment**

The completeness of the vertebrate genomes used for this study was assessed with BUSCO v5.1.2[70] using the vertebrata odb10 database (Supplementary Data 1, Supplementary Table 1, Supplementary Fig. 1), except for three extremely large genomes (Dipnoi: *Protopterus annectens* and *Neoceratodus forsteri*; Amphibia: *Ambystoma mexicanum)*, for which BUSCO results were retrieved from previous studies[71–73]. Since it is expected that genomes with a large proportion of missing BUSCO genes will produce biased estimates for the number of chemoreceptor genes, we only selected high-quality genome assemblies on the basis of two different BUSCO score thresholds: 80% and 90% complete BUSCO genes. In jawed vertebrates, 1532 genome assemblies featured at least 80% complete BUSCO genes (referred to as BUSCO80) and 1181 genome assemblies contained at least 90% complete BUSCO genes (referred to as BUSCO90).

We noticed that this BUSCO filtering strategy was not applicable to jawless fish. All five agnathan genome assemblies had very low BUSCO scores (between 49.5 and 62.4%; Supplementary Fig. 1), despite the fact that three of them are chromosome-level assemblies. We further observed that the same set of 1014 BUSCO genes was

consistently found to be missing or fragmented in these genomes (Supplementary Fig. 2a). To assess if the high number of common missing or fragmented BUSCO genes in agnathans is a true biological result, due to assembly or sequencing artifacts, or due to chance, we performed two rounds of simulations (with 10,000 replications each) in which we randomly extracted N genes, for every agnathan species, in the vertebrata_odb10 database (whereby N was the number of missing/fragmented genes in each species) and then calculated the number of genes that the five species have in common. In one round of simulations the probability of extracting a gene was weighted by gene length (as, in case of assembly artifacts, long genes are more likely to become fragmented or missing than shorter ones). In the other round, all genes had that same probability. The number of common missing genes in both simulations was much lower than the observed number of 1014 genes (Supplementary Fig. 2b), suggesting that the absence of these genes is likely a biological reality. It is also unlikely that these genes are missing in the genome assemblies due to the programmed DNA elimination known to occur in the somatic cells of lamprey and hagfishes[74], as the Reissner and sea lamprey (*Lethenteron reissneri* and *Petromyzon marinus*) assemblies are based on germline sequencing[75,76]. Instead, it seems that the BUSCO gene set is not a suitable quality criterion for jawless fish. We thus decided to include the five agnathan genomes in our analyses, despite their low BUSCO scores.

Finally, we removed one lepidosaur species (*Sceloporus occidentalis*) from our analyses, as we systematically retrieved chemoreceptor subclades from this genome assembly that were not found in any other lepidosaurs genome but instead matched amphibian sequences in the NCBI nr database with high confidence. To further investigate this, we used all available lepidosaur chemoreceptor genes as queries in a BLASTN search against a database composed of all available amphibian chemoreceptor genes and a default e-value of 10. Whenever there was at least one blastn match, we extracted the

lepidosaur query and the amphibian best-hit sequences, translated them to proteins, and aligned them with MAFFT[77]. PAL2NAL[78] was then used to reverse translate these protein alignments into DNA alignments. The function "seqidentity" of the R package 'bio3d'[79] was then used to compute the sequence identity between the lepidosaurs queries and their amphibian best-hits. Chemoreceptor sequences extracted from the *S. occidentalis* genome that did not have orthologues in other lepidosaurs had a much greater sequence identity to amphibian chemoreceptors than any other chemoreceptors (Supplementary Fig. 3).

### Chemoreceptor gene mining

Chemoreceptor genes were mined in all genomes using two different procedures, one adapted for the single-exon gene families, and one for the multi-exon genes. Note that for the *V1R* gene family, both procedures were used, depending on the clade. This is because the *V1R* genes of ray-finned fishes (commonly referred to as *ORA* genes in this group) have several exons, while *V1R* genes of all other species consist of only one exon. The efficiencies of our procedures were assessed by comparing the number of genes retrieved in the same species in previous studies. Although different estimates of the number of genes are expected due to the methodology used (for example, different gene length thresholds, different blast e-values) and different genome assemblies (for which we could not get information in most studies surveyed), we still found a very similar number of chemoreceptors per species (Supplementary Figs. 4 and 5). All scripts implementing these procedures and the required databases are available on GitHub (https://github.com/MaximePolicarpo/Vertebrate_Chemoreceptors_mining, https://doi.org/10.5281/zenodo.10301608).

### Single-exon genes

(*OR*, *TAAR*, *V1R* in non-ray-finned fishes and *T2R*). Known protein sequences from previous studies[19,20,40,80–85] were used as queries in a tblastn[86] search against every genome, with an e-value of 1e−5. Non-overlapping best-hit regions were extracted and extended 1000 bp upstream and downstream using samtools faidx[87]. We then extracted open reading frames (ORFs) present in these regions using EMBOSS getorf[88], with a length threshold of 750 bp for *OR* and *T2R* genes[89,90], or 810 bp and 850 bp for *V1R* and *TAAR* genes (which is a bit lower than the length of the smallest known gene in these families: 820 bp and 870 bp, respectively). As recent studies have shown that some *OR* and *TAAR* genes can also, in rare cases, have two or three exons[84,91,92], we used EXONERATE[93] to search for potential multiple exons in regions in which no ORF was detected. All extracted DNA sequences were then translated into protein sequences with EMBOSS transeq and used as a query in a blastp search against a custom database of GPCR protein sequences. This database was constructed using known chemoreceptor genes and non-chemoreceptor GPCR genes extracted from UniProt[94]. Sequences that best matched to a member of the desired family were then kept and further aligned with known protein sequences of this family (which consist of a representative subset of the sequences used in the initial tblastn search) as well as with outgroup sequences. Outgroup sequences used for each chemoreceptor family can be found in Supplementary Data 1. A maximum-likelihood tree was then computed with IQ-TREE2[95] and sequences that clustered with known chemoreceptor genes were kept and classified as (*i*) 'complete' genes. In order to identify incomplete sequences, these complete genes as well as chemoreceptors from previous studies were used as queries in a second round of tblastn searches against the genome, this time with a more stringent e-value of 1e−20. Again, nonoverlapping best-hit regions were extracted and incomplete gene sequences were predicted in these regions using a combination of tblastn and EXONERATE. These sequences were then used as queries in a blastx search against our custom GPCR database and only sequences that best matched a member of the desired family were retained. These

incomplete sequences were classified in three categories: (*ii*) 'pseudogene', if at least one loss-of-function (LoF) mutation was found; (*iii*) 'truncated', if no LoF was found and if the sequence was not near a contig or scaffold border; or (*iv*) 'edge', if the sequence was close to a contig or scaffold border, which is indicative of an assembly artifact. Finally, Phobius[96] and TMHMM[97] were used to detect the presence of a seven-transmembrane domain typical for GPCR in all complete sequences. All sequences – complete or incomplete – that had at-least one ambiguous nucleotide were classified as 'ambiguous'.

### Multi-exon genes

(*V2R*, *V1R* of ray-finned fishes and *T1R*). Known protein sequences from previous studies[20,39,82,98,99] were used as queries in a tblastn search against each genome, with an e-value of 1e−5. All blast hits were then extended 30,000 bp upstream and downstream and resulting non-overlapping genomic regions were extracted using samtools faidx. EXONERATE was then used to predict chemoreceptor sequences in these regions. In order to avoid extracting a gene prediction that overlapped with two or more real genes, we used an iterative approach to sort EXONERATE results. First, we discarded EXONERATE predictions if the number of exons was higher than the number of exons inferred from tblastn results (that is, the number of non-overlapping tblastn hits that are at least 50 bp long and at least 100 bp distant form each other). Then, we applied a length threshold, keeping only EXONERATE predictions in which the length was equal or higher than the mean expected gene length (900 bp for *V1R* genes and 2700 bp for *V2R* and *T1R* genes). Finally, if two overlapping predictions met these criteria, we kept the one with the best EXONERATE score. We repeated this process decreasing the threshold length until no EXONERATE prediction was found any more. EXONERATE-predicted sequences were then classified into four categories: (*i*) 'complete', if a proper CDS was found with a length of at least 810 bp, 2100 bp and 2200 bp for *V1R*, *V2R* and *T1R* genes, respectively; (*ii*) 'pseudogene', if at least one LoF mutation was detected; (*iii*) 'truncated', if no proper CDS and no LoF mutation was found; or (*iv*) 'edge', if no proper CDS was found and the sequence was close to a contig or scaffold border. The sequences were then translated into protein sequences with EMBOSS transeq (first removing LoF mutations present in pseudogenes) and used as queries in a blastp search against our custom GPCR database. Predictions that best matched a chemoreceptor of the desired family were then aligned with known chemoreceptor proteins and outgroup sequences. A maximum-likelihood tree was computed with IQ-TREE2 and sequences that did not cluster with known chemoreceptor genes were discarded. We also discarded *V2R* and *T1R* sequences smaller than 400 bp, due to difficulties in assigning these to a chemoreceptor family in the light of our blast and phylogeny filtering procedure. In a final step, we used Phobius and TMHMM to detect the presence of a seven-transmembrane domain in the complete sequences. All sequences – complete or incomplete – that had at-least one ambiguous nucleotide were classified as 'ambiguous'.

### Chemoreceptor gene trees and gene delineation

Given the large number of retrieved complete gene sequences in each chemoreceptor family, we first used MAFFT v7.467[77] to generate a template alignment for each gene family, using protein sequences retrieved in previous studies as well as outgroup sequences. We then added all retrieved complete sequences with a seven-transmembrane domain predicted by Phobius and/or TMHMM to these template alignments using the option "-add" in MAFFT. FastTree2[100] with default options was used to infer near maximum likelihood phylogenies from these large alignments, with local support values computed with a Shimodaira-Hasegawa test. Phylogenetic trees were visualized using the R package ggtree[101]. Complete sequences without a predicted seven-transmembrane domain, as well as incomplete (in the categories 'pseudogene', 'truncated' and 'edge') and ambiguous sequences were

classified based on their best blastx match. In order to test for the robustness of the phylogenetic trees computed by the near maximum likelihood method, and to have a clearer view of the different expansions of genes that occurred in vertebrate (sub)classes, we also computed maximum likelihood phylogenies for each gene family. We first extracted all the complete genes belonging to one representative species per vertebrate (sub)class (except for *T1R* genes for which we selected two ray-finned fish species, one teleost and one non-teleost) and aligned those genes using MAFFT v7.467. A maximum likelihood tree was then computed using IQ-TREE2[95] with the optimal model found by ModelFinder[102]. The robustness of the nodes was evaluated with 1000 ultrafast bootstraps[103]. In addition to the phylogenetic evidence, we wanted to confirm that the *T2R* genes found in cartilaginous fish and the *V1R* genes found in birds were also best-matching against known *T2R* and *V1R* genes, respectively. We thus used the complete *T2R* genes of cartilaginous fish as a query in a blastp search against the NCBI nr database and against the NCBI nr database with all cartilaginous fish sequences removed (taxid:7777) (Supplementary Table 5). In a similar way, complete *V1R* genes of birds were used as a query in a blastp search against the NCBI nr database and against the nr database but with all bird sequences removed (taxid:8782) (Supplementary Table 6).

### Ecological and morphological data

(for mammals, birds, ray-finned fishes). Ecological data of mammals were extracted from the EltonTraits[44] and PanTHERIA[45] databases. Ecological data of birds were extracted from AVONET[42] and ecological data on fishes were taken from fishbase[41] (Supplementary Data 1, Supplementary Table 7). Diet preference of mammals and birds were retrieved from the MammalDIET[43] and the AVONET databases, respectively. For ray-finned fishes, diet preferences were inferred from the trophic levels retrieved from fishbase, following their recommendations (https://www.fishbase.se/manual/English/fishbasethe_ecology_table.htm). Thus, species with a trophic level ≤2.19 were classified as herbivores, species with a trophic level ≥2.2 and ≤2.79 were classified as omnivores, and species with a trophic level ≥2.8 were classified as carnivores. We inferred the ancestral diet preference for all branches of the mammalian phylogeny with PastML[104] using the Felsenstein81 model and the MPPA (marginal posterior probabilities approximation) prediction method. Data on the relative olfactory bulb size of mammals and birds, the mean number of turbinates in mammalian olfactory epithelium as well as the mean number of lamellae in the olfactory epithelium of fishes were taken from previous studies[20,53–55] (Supplementary Data 1). Data on the presence/absence of an accessory olfactory bulb in bats were taken from[24–27]. The correlation between the AOB presence/absence in bats and the number of *V1R* genes could only be done with BUSCO80 species, as the removal of *Miniopterus schreibersii* (BUSCO score of 89%) would result in a single clade with the presence of an AOB.

### T1R loss analysis in mammals

We reclassified *T1R* genes of mammals (in species with at least 80% complete BUSCO genes) in four categories: (*i*) 'complete', if the gene was complete in the genome assembly, or if we could retrieve the complete CDS by merging gene fragments ('truncated' and 'edge' sequences), or if the complete CDS was present in another genome assembly of the same species; (*ii*) 'high confidence loss, if the gene was found as a pseudogene in the genome assembly, and if there were at least two distant LoF mutations in the CDS, and/or if the same LoF mutation(s) were found in another genome assembly of the same species, and/or if the same LoF mutation was shared with at-least one closely related species; we also classified as 'high confidence losses' those cases, where the gene was completely missing from an assembly, while the flanking genes were both found and were on the same scaffold, indicating that the *T1R* loss most likely represents a true

deletion; (*iii*) 'low confidence loss', if the gene was found as a pseudogene in the genome assembly but with only one LoF mutation that we could not verify in another genome assembly of the same species; we also classified as 'low confidence losses' those cases, where a gene was completely missing from the genome assembly but where the flanking genes were also not retrieved, or scattered on different scaffolds; (*iv*) 'undetermined', if the gene was initially categorized as 'edge', 'truncated' or 'ambiguous' and if we were not able to retrieve the complete CDS by merging these fragments nor retrieve it in another genome assembly of the same species. Branches on the mammalian phylogeny where *T1R* losses occurred were inferred using the LoF mutations. For example, all cetaceans shared at least one LoF mutation in *T1R1* and *T1R3*, but no common LoF mutation was found in *T1R2*, except between Mysticeti (baleen whales) and between Odontoceti (toothed whales).

To test if, in mammals, carnivore species lost *T1R2* significantly more often than omnivore and/or herbivore ones, we first counted the number of independent *T1R2* losses that occurred on carnivore branches in the mammalian phylogeny as well as the number of carnivore branches with an intact *T1R2* gene (which is equal to the total number of carnivore branches in the tree minus the branches where a *T1R2* loss occurred as well as all their daughter branches). The same was done for omnivore and herbivore branches, as well as for *T1R1* and *T1R3* genes (Supplementary Fig. 37). Count data were the compared with a Chi-squared and Fisher's Exact tests. This strategy is more appropriate than performing a rough count of the number of branches with a *T1R2* loss versus the number of branches with an intact *T1R2*, as it corrects for phylogenetic signal. We then used BayesTraits[105], which allows to test the co-evolution between two binary traits to be tested. Accordingly, we assigned two binary traits to each terminal branch of the tree: *T1R2* complete (1) or pseudogene (0) or undetermined (-); Carnivore (1) or Herbivore/Omnivore (0). We ran BayesTraits with three models: Model1 where the two traits evolve independently; Model2 where both traits co-evolve; Model3 where the *T1R2* state depends on the diet state but not the other way around. For each model, the transition rate of *T1R2* from 0 to 1 (pseudogene to complete gene) was set to 0 and the maximum likelihood algorithm was run 10,000 times to ensure stable results (option MLTries = 10,000). Model2 and Model3 were then compared with Model1 by means of a likelihood ratio test. The same procedure was repeated for *T1R1* and *T1R3*. Finally, to complement the two statistical tests described above, we performed simulations, using the empirical data (25 independent *T1R2* losses in the mammalian phylogeny) as a basis. To do so, we initially assigned a complete *T1R2* gene state to each branch. Then, 25 branches with a complete *T1R2* gene were drawn at random and sequentially, and each time a branch was drawn, this branch and all its daughter branches were assigned a non-functional *T1R2* gene state. If all branches of a tree were assigned a non-functional *T1R2* gene state before the 25 losses could be distributed, the simulation was discarded. We repeated this procedure until 10,000 simulations were performed. Then, the *P*-value for *T1R2* losses in carnivores was defined as the number of simulations where the same number (or more) of independent *T1R2* losses occurred than observed on carnivore branches, divided by the total number of simulations. This was repeated two times: once without considering branch length and once with the drawing probability weighted by branch lengths. The same procedure was followed for *T1R1* and *T1R2* genes, but in these cases adjusting the number of sequential draws to 22 for *T1R1* and to 21 for *T1R3*.

### Selection test on marine tetrapods *OR* genes

In order to detect *OR* genes experiencing accelerated evolution (positive selection or relaxed selection) in marine tetrapods, we used a method initially designed to detect *OR* genes under accelerated evolution in the human lineage[106], with slight modifications. For each marine tetrapod tested (called species$_A$ thereafter), we used a blastx of

all its complete *OR* genes against a database containing all complete *OR* genes of two closely related terrestrial species (named species$_B$ and species$_C$ thereafter). For each species$_A$ *OR* gene, we then extracted the best blastx match of species$_B$ and the best blastx match of species$_C$ and the protein sequences of these three genes were aligned using MAFFT v7.467. This protein alignment was converted into a codon alignment using PAL2NAL[78]. This procedure was repeated for each species$_A$ *OR* gene. For each alignment, we used PAML[28] to compute the ω ratio of these three genes, under five different branch models, and using an unrooted species tree. A one ratio model, called Model 1 in which the three genes evolve under the same ω value. A two-ratio model, called Model 2, where the ω value in species$_A$ is different from the ω value of the other lineages (species$_A$ and species$_B$). Two other two-ratio models were used, Model 3 and Model 4 allowing a different ω value for species$_B$ and for species$_C$ respectively. Finally Model 5, which is a free-ratio model, allows a different ω value per branch. We first assessed which was the best two ratio model by choosing the one with the highest likelihood score. Then this best two-ratio model was compared to the Model 1 conducting a likelihood ratio test and using the χ2 distribution with 1 degree of freedom (df). If this test was not significant, we considered Model 1 as the best model. Otherwise, if the best two-ratio model was significantly better than Model 1, we then compared Model 5 with this best two-ratio model using a second likelihood ratio test (1 df). If this second test was significant, the Model 5 was considered as the best model. Otherwise, the best two-ratio model was retained as the best model. This method was applied to five cetacean species (*Tursiops truncatus, Orcinus orca, Monodon monoceros, Physeter catodon* and *Megaptera novaeangliae*), four pinnipeds (*Mirounga angustirostris, Phoca vitulina, Zalophus californianus* and *Odobenus rosmarus divergens*), two sirenian species (*Dugong dugon* and *Trichechus manatus latirostris*), four sphenisciformes (*Pygoscelis adeliae, Eudyptula minor, Megadyptes antipodes antipodes* and *Eudyptes robustus*), three marine snakes species (*Hydrophis cyanocinctus, Hydrophis curtus* and *Laticauda colubrina*) and three marine turtle species (*Dermochelys coriacea, Caretta caretta, Chelonia mydas*).

### Selection test on bats *V1R* genes and gene tree-species tree reconciliation

All complete *V1R* gene protein sequences of bats for which we had information on the presence or absence of a vomeronasal organ were aligned using MAFFT v7.467. This protein alignment was converted to a codon alignment using PAL2NAL. A maximum likelihood tree was computed from this alignment using IQ-TREE2, with the optimal model detected by ModelFinder. The robustness of the nodes was evaluated with 1000 ultrafast bootstraps. We then used PAML, with the unrooted *V1R* phylogenetic tree to compute maximum likelihood estimates of ω, under three models. A one-ratio model where all the branch evolves under the same ω ratio; A two-ratio model assuming one ω for *V1R* genes belonging to bats lacking an accessory olfactory bulb and one ω for all other branches; A free-ratio model allowing a different ω for each branch. The two-ratio model and the free-ratio model were compared to the one-ratio model conducting a likelihood ratio test, with 1 degree of freedom (df) and 184 df, respectively. The maximum likelihood phylogenetic tree of *V1R* genes was then used to reconstruct the evolutionary dynamic of this gene family in bats. First, nodes with low bootstrap values (<90%) were collapsed into polytomies using the R package ape[69]. Then, we used NOTUNG[107] with the phylogenomics option to root and reconcile this gene tree with the bat species tree.

### Selection test on birds *V1R* gene

To assess if the *V1R* gene copy present in birds was experiencing a different selective force than the *V1R* genes of other tetrapods, protein sequences of complete *V1R* genes found in birds were aligned with closely related *V1R* genes (from the same clade, *V1R7*) from

crocodilians, lepidosaurs, amphibians and testudines, using MAFFT v7.467. PAL2NAL was used to convert this protein alignment into a codon alignment. IQ-TREE2 was then used to compute a maximum likelihood tree, with the optimal model found by ModelFinder. The codon alignment and the phylogenetic tree were then used to compute maximum likelihood estimates of ω with PAML. Five branch models were tested: (*i*) a one-ratio model (Model 1) that assumes that every branch has the same ω ratio; (*ii*) a two-ratio model (Model 2) assuming one ω for bird *V1R* genes and one ω for all other branches; this model is intended to test if V1R genes experienced a shift of ω after the loss of the vomeronasal organ. Because it is still under debate whether the vomeronasal organ is present or not in crocodiles and turtles[9], we used two other two-ratio models to test two alternative hypotheses. (*iii*) Model 3 assumes one ω for the *V1R* genes of birds and crocodiles and one ω for all other branches (testing the hypothesis of vomeronasal organ loss in the MRCA of crocodiles and birds); (*iv*) Model 4 assumes one ω for the *V1R* genes of birds, crocodiles and turtles and one ω for all other branches (testing the hypothesis of vomeronasal organ loss in the MRCA of crocodiles, birds and turtles). (*v*) finally, we also ran a free-ratio model, which allows a different ω value per branch. The best two-ratio model was chosen based on the highest likelihood value, and this two-ratio model was compared to the one-ratio model by means of a likelihood ratio test based on the χ2 distribution with 1 df. Then, another likelihood ratio test was used to compare the free-ratio model and the best two-ratio model (75 df). Given the shift of bird *V1R* genes toward an ω value of 1, we then used RELAX[29], implemented in the HyPhy software[108], to decipher if these genes were evolving under positive selection or relaxed selection. RELAX uses a branch-site model and test for the convergence of a distribution of three ω values toward 1 in a lineage (named "test" lineage). The magnitude of this convergence depends on K, which tends to 0 as selection tends to complete relaxation. Thus, the bird *V1R7* clade was assigned as test lineage, while all other branches were assigned as foreground.

### Selection test on marine mammals T1R genes

We assessed the selective forces acting on sirenian *T1R1, T1R2* and *T1R3* genes, as sirenians are the only marine mammals having a complete *T1R* gene repertoire. We first used MAFFT v7.467to align the protein sequences of sirenian *T1R1* genes with protein sequences of pinniped and cetacean *T1R1* pseudogenes (by first removing stop codons and frameshifts present in these pseudogene sequences), as well as with *T1R1* protein sequences of terrestrial species closely related to these three marine clades. We then used PAL2NAL to convert this protein alignment into a codon alignment. This codon alignment and the mammal species tree were then used to compute maximum likelihood estimates of ω with PAML. Five branch models were used: (*i*) a free-ratio model (Model 1), which allows a different ω value per branch; (*ii*) a two-ratio model (Model 2) assuming one ω for marine branches and one ω for terrestrial branches; (*iii*) a two-ratio model (Model 3) assuming one ω for cetacean and pinnipeds branches and one ω for terrestrial and sirenian branches; (*iv*) a three-ratio model (Model 4), assuming one ω for cetacean and pinniped branches, one ω for sirenian branches, and one ω for terrestrial branches; and (*v*) a null model assuming that every branch has the same ω ratio. This procedure was repeated for *T1R2* and *T1R3*. For the three *T1R* genes, the best model was always the free-ratio model (Model 1). The choice between Model 2 and Model 3 was always done based on the highest likelihood score, and we systematically compared Model 4 with Model 2 and Model 3 by the mean of a likelihood ratio test, using the χ2 distribution with 1 df. When Model 4 was better than both Model 2 and Model 3 (*T1R1* and *T1R3*), then it was assumed that the ω ratio was significantly different between sirenians, terrestrial species and pinnipeds/cetaceans. For *T1R2*, Model 4 was significantly better than Model 2, but was not significantly better than the Model 3, which we interpreted as the ω ratio

being not significantly different between terrestrial species and sirenians, but significantly different from the ω ratio of pinnipeds and cetaceans.

Finally, for the three *T1R* genes, we used RELAX to examine if any sign of positive selection or relaxation of selection could be found in sirenians. Terrestrial branches were assigned as foreground branches, while cetaceans, pinnipeds and sirenians were sequentially assigned as "test" branches in three runs of RELAX (and for the three *T1R* genes). Note that RELAX successfully detect a relaxation of selection on cetaceans and pinnipeds for the three *T1R* genes, coherent with the fact that these genes were found with loss-of-function mutations in these species.

## Phylogenetic comparative analyses

The function pgls of the R package caper[109] was used to perform all phylogenetic generalized linear models presented in this study. Phylogenetic tree manipulations were done with the R packages *phytools*[110] and *ape*[69]. The graphical representation of phylogenetic trees were done with the R package ggtree[101]. All other plots were done with *ggplot*[111]. Animal silhouettes used in this study were retrieved from http://phylopic.org/ (full links for each silhouette can be found in Supplementary Data 1).

## Impact of species trees topologies on pGLS

We assessed the impact of the species phylogenies on the pGLS performed in this study. For mammals, we first extracted the 1000 most represented BUSCO genes only considering species with at least 80% complete BUSCO genes. For each gene, a protein sequence alignment was made using MAFFT and the alignment was trimmed using trimAl[112] and the option "-automated1". We then generated fifty concatenated alignments, taking twenty of the single gene alignments at random without replacement, using AMAS[113]. A maximum likelihood phylogeny was computed from each concatenated alignment, using IQ-TREE2 and the LG + F + G4 model. Finally, the least square dating method was used to calibrate these trees using three calibration dates retrieved from TimeTree.org (Supplementary Data 1). Each tree was used to recompute pGLS, and we also computed its Robinson–Foulds distance from the reference tree used in the study using phangorn[114]. The same procedure was followed for birds and actinopterygians. We found that the species tree topologies had very little or no impact on the results (Supplementary Figs. 54–56).

## Reporting summary

Further information on research design is available in the Nature Portfolio Reporting Summary linked to this article.

## Data availability

All data and results from this study are available on FigShare (https://doi.org/10.6084/m9.figshare.c.6972150.v1). This includes nucleotide sequences of all chemoreceptor sequences (fasta) and their clades (txt); chemoreceptor alignments (fasta); chemoreceptor phylogenetic trees (newick); all species phylogenetic trees (newick) used in this study; PastML results for mammals diet preferences; fifty random concatenated alignments (fasta); and calibrated species trees (newick) for mammals, birds and ray-finned fishes. Data extracted from EltonTraits, PanTHERIA, AVONET, Fishbase and MammalDIET are available in Supplementary Data 1. Accession codes for all genome assemblies analyzed in this study are available in Supplementary Data 1. Source data are provided with this paper.

## Code availability

All scripts necessary to reproduce the results of this study are available on GitHub (https://github.com/MaximePolicarpo/Vertebrate_Chemoreceptors_mining), https://doi.org/10.5281/zenodo.10301608.)

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

## Acknowledgements

We would like to thank the members of the Salzburger lab and the anonymous reviewers for valuable suggestions and comments on this study. All calculations were performed at sciCORE (http://scicore. unibas.ch/), the center of scientific computing at University of Basel (with support by the SIB/Swiss Institute of Bioinformatics). This work was funded by the Swiss National Science Foundation (SNSF; grants 189970 and 208002) to W.S.

## Author contributions

M.P. and W.S. designed this study, with input from M.W.B. and D.C. M.P. performed all data analyses. M.P. and W.S. wrote the manuscript with input from all authors.

## Competing interests

The authors declare no competing interests.
