## [Peer Review File · Nature Communications]

Diversity and evolution of the vertebrate chemoreceptor gene repertoireThis manuscript has been previously reviewed at another journal that is not operating a transparent peer review scheme. This document only contains reviewer comments and rebuttal letters for versions considered at *Nature Communications*.

Reviewers' Comments:

Reviewer #1:

Remarks to the Author:

I acknowledge the great effort that the authors have put through the revision of this manuscript. Unfortunately, one of main criticism (that is, the lack of functional validation of bitter or sweet genes) from the first round of review wasn't really taken into account.

Reviewer #2:

Remarks to the Author:

The authors present a comprehensive study of chemosensory receptor gene numbers (OR, TAAR, V1R, V2R, T1R, T2R) in over thousand vertebrate genomes. The methods for identifying the receptor repertoires are solid. Several of the results stated in the abstract such as dynamic evolution, lineage-specific expansions and losses of chemosensory receptor gene families are not new and have in fact been reported in many previous publications. However, the clear advance of the present study is that now the conclusions are based on a much larger dataset. Furthermore, the enlarged data set allowed more fine-grained correlation with ecological parameters than previously possible. These correlations have now been included in a new Figure 4, which is a clear improvement above the original version of this manuscript. The correlation coefficients are often relatively small, but that may be expected in large scale species sampling. Overall the revisions have improved the manuscript considerably. It does contain a wealth of data points and as such constitutes an important addition to the knowledge about chemosensory receptors.

The manuscript is generally not easy to read due to the listing of many numbers, genes and Latin species names in the main text. This has not changed from the original version. I would strongly suggest to put e.g. the association values (correlation coefficient, p values) in a dedicated table, this would increase readability of the corresponding segment considerably.

The newly introduced text about correlations between ecology and gene number is a welcome improvement. However it could be better integrated. For example, in new text line 319-337 the authors check the correlations on three levels, BUSCO80 set of genomes, BUSCO90 set and set of chromosomal level assemblies. The same procedure should also be used for the original version text, e.g. line 349-360. Alternatively all such discussions could go to supplementary material, and the differences between the three levels only mentioned summarily. Such a summarily mention is also missing for the respective pGLS discussions (albeit it is mentioned in the rebuttal).

Minor points:

line 44

Reference 10 does not mention lobe-finned fishes.

Line 286

omega ratio should be explained at first occurrence in the main text, not just in figure legend of SI 30.

line 311

'as not true' seems to be a typo.

Lines 319-337

Here the correlation coefficients should be given also, as was done below, line 352ff. Also all these values should go into a table, not be scattered in the text.

line 448-450 Rephrase to make clear that you are talking about T1R loss in carnivore marine mammals. Sirenians are (sort of) marine mammals and herbivores, but here it is claimed that marine mammals are all carnivores.

Abstract, „examination of 2,210 vertebrate genomes“

This number is different from the BUSCO80 number (1423 genomes). Please clarify.

Line 453

The grammar of this sentence is broken.

Figure 4 panel a

The borders of the rectangles are partially obscuring the text.

Reviewer #3:

Remarks to the Author:

The authors have satisfactorily addressed my concerns, including further validation of their species sampling (e.g., chromosome-scale genomes) as well as an in-depth selection analysis. I find the manuscript substantially improved and have no further criticism.

REVIEWERS' COMMENTS

Reviewer #1 (Remarks to the Author):

I acknowledge the great effort that the authors have put through the revision of this manuscript. Unfortunately, one of main criticism (that is, the lack of functional validation of bitter or sweet genes) from the first round of review wasn't really taken into account.

We thank this Reviewer very much for acknowledging our effort. We absolutely agree with this Reviewer that having functional validations for bitter and/or sweet receptor genes would be great. At the same time, we believe that such a massive endeavor is way beyond the focus of the current study.

Reviewer #2 (Remarks to the Author):

The authors present a comprehensive study of chemosensory receptor gene numbers (OR, TAAR, V1R, V2R, T1R, T2R) in over thousand vertebrate genomes. The methods for identifying the receptor repertoires are solid. Several of the results stated in the abstract such as dynamic evolution, lineage-specific expansions and losses of chemosensory receptor gene families are not new and have in fact been reported in many previous publications. However, the clear advance of the present study is that now the conclusions are based on a much larger dataset. Furthermore, the enlarged data set allowed more fine-grained correlation with ecological parameters than previously possible. These correlations have now been included in a new Figure 4, which is a clear improvement above the original version of this manuscript. The correlation coefficients are often relatively small, but that may be expected in large scale species sampling. Overall the revisions have improved the manuscript considerably. It does contain a wealth of data points and as such constitutes an important addition to the knowledge about chemosensory receptors.

The manuscript is generally not easy to read due to the listing of many numbers, genes and Latin species names in the main text. This has not changed from the original version. I would strongly suggest to put e.g. the association values (correlation coefficient, p values) in a dedicated table, this would increase readability of the corresponding segment considerably.

We thank the reviewer for this suggestion. We added a table that summarizes the associations that we found between chemoreceptors and ecological factors.

The newly introduced text about correlations between ecology and gene number is a welcome improvement. However it could be better integrated. For example, in new text line 319-337 the authors check the correlations on three levels, BUSCO80 set of genomes, BUSCO90 set and set of chromosomal level assemblies. The same procedure should also be used for the original version text, e.g. line 349-360. Alternatively all such discussions could go to supplementary material, and the differences between the three levels only mentioned summarily. Such a summarily mention is also missing for the respective pGLS discussions (albeit it is mentioned in the rebuttal).

We thank the reviewer for this suggestion. In the revised version, we added the associations between the number of chemoreceptors and the olfactory organ morphology, taking only chromosome-scale assemblies, to the Supplementary Figure 43 (now Supplementary Figure 53) and added this sentence to the manuscript: "These associations with olfactory organ morphologies hold true when considering the BUSCO90 or chromosome-scale assemblies datasets (Supplementary Fig. 53)". We also added a panel (c) to the Supplementary Figure 6 (now Supplementary Figure 7) showing the correlations between the number of genes in each chemoreceptor family, when only considering chromosome-scale assemblies, which was previously only shown in the responses to reviewers.

Minor points:

line 44

Reference 10 does not mention lobe-finned fishes.

Thank you for detecting this error. We changed the sentence according to the reference: “*V1R* and *V2R* genes are expressed in the sensory epithelium of the vomeronasal organ in tetrapods (except in amphibian, where *V1R* and a subset of *V2R* genes are expressed in the main olfactory epithelium), while in cartilaginous and ray-finned fishes these genes – often referred to as ORA and OlfC in these clades – are expressed in the main olfactory epithelium”

Line 286

omega ratio should be explained at first occurrence in the main text, not just in figure legend of SI 30.

We added “(that is, dN/dS)” after the first occurrence of ω ratio at line 286.

line 311

'as not true' seems to be a typo.

Changed to “is not true”.

Lines 319-337

Here the correlation coefficients should be given also, as was done below, line 352ff. Also all these values should go into a table, not be scattered in the text.

Thank you for this suggestion. We added a table to the manuscript for the ecological associations discussed.

line 448-450 Rephrase to make clear that you are talking about T1R loss in carnivore marine mammals. Sirenians are (sort of) marine mammals and herbivores, but here it is claimed that marine mammals are all carnivores.

Modified.

Abstract, „examination of 2,210 vertebrate genomes“

This number is different from the BUSCO80 number (1423 genomes). Please clarify.

2,210 is the number of genomes that was initially investigated, before any filter of the quality of these genomes. We thus replaced 2,210 with the number of genomes in the smallest dataset (BUSCO80, N=1,527). 1,423 correspond to the number of genomes in the BUSCO80 dataset, for which we retrieved phylogenetic information, and for which we could perform pGLS analysis presented in the study.

Line 453

The grammar of this sentence is broken.

Corrected.

Figure 4 panel a

The borders of the rectangles are partially obscuring the text.

We modified figure 4a for the species label to be visible. Furthermore, we modified the species names so that they are now in italic, and we replaced the underscore between the genus name and the species

specific name by a space.

Reviewer #3 (Remarks to the Author):

The authors have satisfactorily addressed my concerns, including further validation of their species sampling (e.g., chromosome-scale genomes) as well as an in-depth selection analysis. I find the manuscript substantially improved and have no further criticism.

Thank you very much for your suggestions that helped improving the manuscript.